# In-Situ Sludge Reduction Performance and Mechanism in Sulfidogenic Anoxic–Oxic–Anoxic Membrane Bioreactors

**DOI:** 10.3390/membranes12090865

**Published:** 2022-09-08

**Authors:** Chengyue Li, Tahir Maqbool, Hongyu Kang, Zhenghua Zhang

**Affiliations:** 1Institute of Environmental Engineering & Nano-Technology, Tsinghua Shenzhen International Graduate School, Tsinghua University, Shenzhen 518055, China; 2Guangdong Provincial Engineering Research Centre for Urban Water Recycling and Environmental Safety, Tsinghua Shenzhen International Graduate School, Tsinghua University, Shenzhen 518055, China; 3School of Environment, Tsinghua University, Beijing 100084, China

**Keywords:** membrane bioreactor, sludge reduction, sulfide inhibition, microbial community, metabolic pathways

## Abstract

The excess sludge generated from the activated sludge process remains a big issue. Sustainable approaches that achieve in situ sludge reduction with satisfactory effluent quality deserve attention. This study explored the sludge reduction performance of sulfidogenic anoxic–oxic–anoxic (AOA) membrane bioreactors. The dynamics of the microbial community and metabolic pathways were further analyzed to elucidate the internal mechanism of sludge reduction. Compared with the conventional anoxic–oxic–oxic membrane bioreactor (MBR_control_), AOA_S150_ (150 mg/L SO_4_^2−^ in the membrane tank) and AOA_S300_ (300 mg/L SO_4_^2−^ in the membrane tank) reduced biomass production by 40.39% and 47.45%, respectively. The sulfide reduced from sulfate could enhance the sludge decay rate and decrease sludge production. Extracellular polymeric substances (EPSs) destruction and aerobic lysis contributed to sludge reduction in AOA bioreactors. The relative abundance of Bacteroidetes (phylum), sulfate-reducing bacteria (SRB, genus), and *Ignavibacterium* (genus) increased in AOA bioreactors compared with MBR_control_. Our metagenomic analysis indicated that the total enzyme-encoding genes involved in glycolysis, denitrification, and sulfate-reduction processes decreased over time in AOA_S300_ and were lower in AOA_S300_ than AOA_S150_ at the final stage of operation. The excess accumulation of sulfide in AOA_S300_ may inactive the functional bacteria, and sulfide inhibition induced sludge reduction.

## 1. Introduction

The activated sludge process is environmentally friendly, easy to operate, efficient, and economical, and it plays a significant role in sewage treatment [1,2]. Biological treatment uses microbes to convert organics to bacterial cells or cell tissues, carbon dioxide, water, etc. [3,4]. Meanwhile, a large volume of excess sludge is generated, which is predicted to increase, considering the rapid growth of population and urbanization [5]. Various hazardous substances exist in waste activated sludge, such as pathogens, persist pollutants, and heavy metals, which may pose a threat to the environment and public health [6,7]. Incineration, landfill, anaerobic digestion (AD), dewatering, composting, thickening, and drying are conventional sludge treatments, but the improper management of excess sludge could result in secondary pollution [6,8]. Moreover, the cost of sludge handling and disposal accounts for 25–65% of the general operational costs of a sewage treatment plant [9]. Therefore, it is crucial to find sustainable and cost-effective approaches for sludge treatment and disposal [6].

Sludge minimization that affects microbial metabolism without additional resources or energy could be applied as an alternative method for excess sludge disposal and is a green approach to realizing sustainable wastewater treatment [10]. Many strategies based on microbial metabolisms, i.e., electron transport, thermodynamics, and substance metabolism, have been proposed for sludge reduction. The anaerobic side-stream reactor (ASSR) process is a promising sludge reduction strategy, mainly based on cryptic growth, a category of carbon metabolism [10]. The ASSR process could accomplish effective sludge reduction with efficient pollutant removal and refined sludge’s settling ability [11]. It has been reported that membrane bioreactor (MBR)–ASSR, which inserts an ASSR in the sludge return loop of the MBR, could reduce sludge generation by 6.0–61.6% under different operational parameters of ASSRs [12,13,14]. However, the external ASSR increased the footprint of the whole system [15].

A variety of strategies, such as packing carriers [16], ultrasonication [17], and introducing influent to ASSR [15], have been applied to further improve the sludge reduction efficiency. Sulfidogenic bioaugmentation is also regarded as a useful approach for sludge minimization. Huang et al. [18] confirmed that the sulfidogenic oxic-settling anaerobic (SOSA) process could reduce sludge production by 57%, with efficient pollutants’ removal. After sulfate was added to the anaerobic side-stream reactor (6 g S/m^3^_sewage_), sulfate-reducing bacteria (SRB), the slow grower, was enriched. Sulfate was reduced by SRB, while organics were removed. The sulfide reduced from sulfate by SRB could induce sludge disintegration and enhance the sludge decay rate. Meanwhile, functional microorganisms would be inactivated by sulfide [18,19,20]. As such, the sludge production could be reduced. Huang et al. [21] determined that coupling an electrochemical pretreatment (EPT)-integrated SOSA (named ESSR) with the conventional activated sludge (CAS) process reduced sludge by 61%, with refined effluent quality. Li et al. [22] evaluated the economic and environmental effects of the SOSA process and found that it could decrease the cost by 18% and greenhouse gas emissions by 23% compared with the conventional anoxic–oxic (AO) process. It is necessary to expand the application of the SOSA system, such as MBR–SOSA [18]. Li et al. [23] confirmed that anoxic–oxic–anoxic (AOA) bioreactors could achieve efficient sludge reduction (i.e., 30%). It is meaningful to couple the sulfidogenic bioprocess with the AOA bioreactor to further reduce sludge generation to achieve more economic benefits.

These approaches achieve efficient sludge reduction without influencing effluent quality by regulating enzymatic activities and screening functional microorganisms, which affects the microbial community dynamics [24]. Zhou et al. [25] verified that the growth of hydrolytic and fermentative bacteria (class level—*Actinobacteria* and *Anaerolineae*; genus level—*Sulfuritalea*, *Propionivibrio*, and *Azospira*), as well as slow-growing bacteria, *Trichococcus*, was accelerated after the insertion of a sludge holding tank and contributed to sludge reduction. Ferrentino et al. [26] highlighted that slow growers, i.e., denitrifying phosphate-accumulating organisms (DPAOs), and SRB are important to sludge reduction. Huang et al. [18] also pointed out that the SOSA process reduced sludge generation, since sulfidogenesis enhanced the growth of slow growers, such as sulfide-oxidizing bacteria (SOB), SRB, and hydrolytic/fermentative microorganisms. However, alternating the operational parameters not only affected the microbial-community-structure dynamics but also the microbial metabolism. Metagenomics is a useful approach to assess the microbial community by taxonomic and functional profiling [27]. Li et al. [23] applied a metagenomic analysis to compare differences in relevant genes involved in the carbon and nitrogen metabolic processes between the conventional membrane bioreactor and AOA bioreactors. However, the identification of variations in functional genes in the sulfidogenic AOA processes is necessary to determine the underlying mechanism. In addition, the intrinsic association between microbial metabolism and sludge reduction in the sulfidogenic AOA processes remains unknown.

This study evaluated the in situ sludge reduction performance of sulfidogenic anoxic–oxic–anoxic membrane bioreactors. Sulfate solutions were directly pumped into the anoxic membrane tank of AOA bioreactors. The objectives of this project were as follows: (i) to compare the effect of sulfate on nutrient removal, sludge reduction, and EPS characteristics in AOA bioreactors; and (ii) to investigate the sludge reduction mechanisms by 16S rRNA amplicon sequencing and metagenomics.

## 2. Materials and Methods

### 2.1. Experimental Setup

Three lab-scale bioreactors were operated in parallel. The conventional MBR (anoxic–oxic–oxic) was used as the control (MBR_control_). Two sulfidogenic AOA bioreactors were named as AOA_S150_ (150 mg/L SO_4_^2−^ in the membrane tank) and AOA_S300_ (300 mg/L SO_4_^2−^ in the membrane tank), respectively. Eighty percent of the sludge was refluxed from the membrane tank to the pre-anoxic tank in the AOA_S150_ and AOA_S300_ bioreactors (23 h/day), and the recirculation ration increased by up to 1600% (1 h/day) [23]. The schemes of three bioreactors are shown in Figure 1. All the bioreactors consisted of three tanks (anoxic tank (DO < 0.5 mg/L), 2 L; oxic tank (DO of 1.5–2.5 mg/L), 3.2 L; and membrane tank, 2.2 L), and each tank was separated by plexiglass. MBR_control_ had an air pump in the membrane tank (DO of 1.5–2.5 mg/L), while stirs were used to keep the sludge mixed well and created a low-oxygen environment in the membrane tank of AOA bioreactors (DO < 0.5 mg/L). Different from the gravity-flow mixing model in MBR_control_, mixed liquor shifting in AOA bioreactors was controlled by peristaltic pumps from the anoxic to oxic to membrane tanks. Flat Al_2_O_3_ ceramic membrane modules (Meidensha, Tokyo, Japan) with an 0.1 µm average pore size and 0.0425 m^2^ effective area were put in the membrane tanks of three bioreactors. The operating parameters of flux, hydraulic retention time (HRT), and sludge retention time (SRT) were 5 LMH, 35 h, and 80 days, respectively. The sludge recirculation rate from the oxic to pre-anoxic tank was the same in all three bioreactors (200%).

Synthetic wastewater was used as feed for all three bioreactors, and the detailed compositions are listed in Appendix A. In total, 792 mL extra sulfate solution was pumped into the AOA bioreactors. To offset the dilution effect of the additional sulfate solution, the AOA bioreactor feed was concentrated accordingly. The entire operating period comprised a 2-month acclimatization phase and a 3-month stable phase.

### 2.2. Analytical Characterization

Concentrations of dissolved organic carbon (DOC), total dissolved nitrogen (TN), ammonium (NH_4_^+^), nitrate (NO_3_^−^), total phosphate (TP), and mixed liquor suspended solids (MLSS) were determined 1 or 2 times per week, following APHA standard methods [28]. The extraction of soluble microbial products (SMPs) and extracellular polymeric substances (EPSs) was conducted by following Zhang et al. [29]. The measurements of polysaccharide and protein content in SMPs and EPSs were performed by phenol/H_2_SO_4_ and modified Lowry methods, respectively [29,30,31]. A one-way factor analysis of variance (ANOVA) was applied to test the significance of the correlation analysis.

### 2.3. 16S rRNA Amplicon Sequencing and Metagenomics

The 27 sludge samples gathered from the anoxic, oxic, and membrane tank of three bioreactors (operating days 1, 45, and 90) were used for 16S rRNA amplicon sequencing. Following the DNA extraction, PCR amplification and product sequencing were performed at MAGIGENE Bioinformatics Technology Co. Ltd. (Guangzhou, China). Microbial community analysis was conducted according to Ren et al. [32].

Meanwhile, sludge samples in the membrane tank of AOA_S150_ and AOA_S300_ bioreactors (on days 1, 45, and 90) were collected for metagenomic analysis. Six samples were obtained for DNA extraction, and the Illumina Miseq 2500 platform was applied for metagenomic sequencing analysis after library preparation [23]. Trimmomactic software (Bjoern Usadel, Jülich, Germany) (v 0.32) was applied for quality control [33], and then the clean reads were assembled by the MEGAHIT software (Tak-Wah Lam, Tokyo, Japan) (v 1.0.6) (k-min 35, k-max 95, k-step 20) to acquire the assembly scaftigs [34]. The Prodigal program (Doug Hyatt, Knoxville, TN, USA) (v 2.6.3) was run to predict protein-coding genes from genomic data [35]. Gene clusters were predicted by Linclust [36]. The Unigenes Catalog was then created for further analysis. To determine the gene abundance, clean reads were mapped to the Unigenes Catalog, using bbmap (https://jgi.doe.gov/data-and-tools/software-tools/bbtools/, accessed on 29 September 2022). The glycolysis (map00010), nitrogen metabolism (map00910), and sulfate reduction pathway (map00920) diagrams were determined from the Kyoto Encyclopedia of Genes and Genomes (KEGG) database [37].

## 3. Results and Discussion

### 3.1. Basic Water-Quality Parameters

Temporal and average variations in basic quality parameters (DOC, TN, and TP) in the effluents of three bioreactors (stable phase) are shown in Appendix A. More than 98% of organic substrates were removed by membrane filtration and the high heterotrophic and autotrophic activities in MBRs [38,39]. The TP concentration in the effluent of three reactors failed to meet the discharge standard in China (<0.5 mg/L). The relatively poor TP removal efficiency (>81% for MBR_control_, >76% for AOA_S150_ and AOA_S300_) might be due to the long SRT in the three bioreactors [17]. However, AOA_S150_ and AOA_S300_ presented higher TN removal efficiency (>93%) than the MBR_control_ (>81%).

After sulfate was added, the variations in the NH_4_^+^-N and NO_3_^−^-N concentrations in the anoxic, oxic, and membrane tanks were monitored, and they are shown in Appendix A. In MBR_control_, NH_4_^+^-N decreased gradually from the anoxic (9.35 ± 3.68 mg/L) to the oxic (0.47 ± 0.62 mg/L) to the membrane tank (0.11 ± 0.24 mg/L), while NO_3_^−^-N increased from the anoxic (0.16 ± 0.30 mg/L) to the oxic (4.60 ± 2.36 mg/L) to the membrane tank (5.39 ± 2.27 mg/L), highlighting the role of nitrification in NH_4_^+^-N removal. However, in the AOA_S150_ and AOA_S300_ bioreactors, NH_4_^+^-N increased from the oxic (0.28 ± 0.33 and 0.47 ± 0.43 mg/L) to the membrane tank (0.66 ± 0.81 and 0.94 ± 1.60 mg/L) because ammonification and sludge disintegration occurred in the membrane tank [25]. The release of NH_4_^+^-N indicates sludge reduction [40]. In the anoxic membrane tank, organics released from sludge decay could be utilized as a carbon source for denitrification [18,25]. Moreover, the NO_3_^−^-N concentration in the membrane tank of the AOA_S150_ and AOA_S300_ bioreactors was 0.78 ± 0.75 mg/L and 0.60 ± 1.11 mg/L, respectively, lower than that of the MBR_control_. This was because the sulfide reduced from sulfate by the SRB could work as the electron donor for sulfur autotrophic denitrification to decrease the NO_3_^−^-N concentration [18].

### 3.2. Sludge Reduction

The variations in the MLSS concentration during the acclimation (60 days) and stable periods (90 days) are shown in Figure 2a. The MLSS concentrations in the different tanks of the MBR_control_, AOA_S150_, and AOA_S300_ bioreactors are shown in Appendix A. The membrane tank of the AOA bioreactors exhibited higher MLSS concentrations than the pre-anoxic and oxic tanks (*p* < 0.01) did. In the MBR_control_, there is no significant difference in the MLSS concentration among the three tanks (*p* > 0.05). Since the sludge recirculation rate from the membrane tank to the pre-anoxic tank in the AOA bioreactors was only 80%, sludge gradually accumulated in the membrane tank. Similar conclusions were presented by Huang et al. [18], in which the side-stream bioreactor also had a higher MLSS concentration than the mainstream bioreactor. In total, the average MLSS concentration in the stable phase was 4.11 ± 0.43, 2.45 ± 0.37, and 2.16 ± 0.34 g/L, respectively. The sulfur bioaugmentation conditions achieved a 40.39% and 47.45% MLSS concentration reduction in the AOA_S150_ and AOA_S300_ bioreactors, respectively, compared with the MBR_control_.

The observed yield coefficient (Y_obs_) was calculated from the MLSS concentration during the stable period of operation, following the method of Ferrentino et al. [26], and it is shown in Figure 2b. The Y_obs_ of AOA_S150_ and AOA_S300_ decreased to 0.164 and 0.146 g MLSS/g DOC, respectively, 43.64% and 49.83% lower than MBR_control_ (Y_obs_ = 0.291 g MLSS/g DOC). The high sulfate dose contributed to the higher sludge-reduction efficiency. Sulfate was reduced to sulfide by SRB in the anoxic membrane tank. The sulfide promoted the endogenous decay and contributed to sludge disintegration [41,42]. Hence, almost 50% less sludge was generated in the AOA_S300_ bioreactor.

### 3.3. SMP and EPS Dynamics in AOA Bioreactors

EPSs (soluble and bound) exhibit various functional properties, including adhesion, aggregation, cohesion, and sorption, as well as being an energy and nutrient source [43]. The variations in EPS components could serve as alternative indicators for evaluating sludge reduction performance [44,45]. In this study, the effects of sulfur-cycle augmentation on EPSs were investigated.

The average concentrations of protein and polysaccharide in the SMP of the three bioreactors are shown in Appendix A. In the MBR_control_, the total SMP concentration decreased from the pre-anoxic tank to membrane tank, and the concentration of protein vs. polysaccharide in the membrane tank was 3.01 ± 0.77 vs. 1.40 ± 1.73 mg/L. In the AOA_S150_ and AOA_S300_ bioreactors, the concentration of protein vs. polysaccharide in the membrane tank was higher than it was for the MBR_control_: 3.91 ± 0.63 vs. 2.17 ± 3.19 mg/L and 4.09 ± 0.76 vs. 2.55 ± 2.54 mg/L. Cell lysis and sludge disintegration induced by sulfidogenesis led to the accumulation of SMPs [42,46], and more SMPs would be generated in response to sulfate stress [47].

Figure 3 shows the average protein and polysaccharide concentrations in the EPS in the three bioreactors. The concentrations of protein vs. polysaccharides in the membrane tank of the MBR_control_ was 18.37 ± 3.45 vs. 7.55 ± 2.99 mg/g MLSS. There was a lower concentration of protein vs. polysaccharides (15.02 ± 6.03 vs. 6.22 ± 2.65 mg/g MLSS) in the membrane tank of AOA_S150_, compared with AOA_S300_, in which the protein vs. polysaccharides concentration was 18.91 ± 4.61 vs. 8.45 ± 2.20 mg/g MLSS. Sulfide might kill microbes [48]; therefore, the higher EPS concentration might be due to high numbers of dead cells. Moreover, in the AOA bioreactors, the EPS concentration was lower in the membrane tank than in the oxic tank. Sulfide induced cell breakage and enhanced EPS degradation, and the released dissolved organic matter (DOM) could be used as the secondary substrate for microbial cryptic growth [21].

Cell lysis and particulate organic matter (POM) hydrolysis during sludge reduction could induce changes of DOM in bioreactors [49]. Studying the altered DOM compositions in different treatment units of three bioreactors was helpful to understand how the sludge reduction mechanism was affected by the sulfur-cycle bioprocess. Excitation-emission matrix–parallel factor (EEM–PARAFAC) analysis was used to identify the DOM characteristics (Appendix A). Maps of three fluorescent components with their emission and excitation loadings are shown in Appendix A. The distribution of three components in SMP and EPS from anoxic, oxic, and membrane tanks of the three bioreactors are shown in Appendix A. Humic-like substance (C1) was more abundant in SMP from the membrane tank of the MBR_control_, while a higher proportion of tryptophan-like substance (C2) than C1 was found in AOA bioreactors. In addition, the total fluorescence intensity (sum of F_max_) decreased from anoxic to oxic to membrane tanks of MBR_control_, while it increased in the membrane tanks of AOA_S150_ and AOA_S300_ bioreactors. The enhanced total fluorescence intensity in the membrane tank of the AOA bioreactors could be because cell lysis and POM hydrolysis occurred during in situ sludge reduction [15]. For the EPS-related fluorescent substance (R.U./g MLSS), there was a gradual decrease in fluorescent substance intensity from the pre-anoxic tank to the membrane tank in the AOA_S150_ bioreactor, while cell lysis occurred in the oxic tank of the AOA_S300_ bioreactor with increased fluorescent substance intensity. The fluorescent intensities in the membrane tanks of the MBR_control_, AOA_S150_, and AOA_S300_ bioreactors were 7.34 ± 1.81, 5.20 ± 2.03, and 6.87 ± 1.85 R.U./MLSS, respectively. The fluorescent intensities in the membrane tank of AOA bioreactors were much lower than the fluorescent intensity in the MBR_control_. The reduced EPS-related fluorescent intensities in the membrane tank of the AOA bioreactors could be due to EPS destruction, which plays an important role in sludge reduction [15,23].

### 3.4. Microbial Community

The alteration of DOM characteristics induced by sludge lysis and hydrolysis has an impact on the microbial community [50,51]. Comparing the microbial community among three bioreactors during the operation revealed the underlying sludge-reduction mechanisms. The evolution of the microbial community was monitored and is presented in Appendix A and Figure 4a,b. The lowest Chao1 values in m_9_ (the membrane tank of AOAS_300_ on day 90) implied the reduction in microbial-community richness (Figure 4a) [52]. In addition, the highest Simpson and lowest Shannon values in m_9_ show that this tank was less diverse than the membrane tank of other bioreactors [53]. The high sulfate concentration reduces the microbial community’s richness and diversity, and this might be due to the competition between SRB and other anaerobes under low-COD/SO_4_^2−^ conditions [21].

The microbial composition was classified at the phylum level, as shown in Figure 4a. The addition of sulfate had a significant impact on the microbial community. The dominant phyla in each reactor were Proteobacteria and Bacteroidetes. After 90 days of operation, Proteobacteria was enriched in MBR_control_ (61.08–67.41%) but decreased in AOA_S150_ (44.41–63.35%) and AOA_S300_ (37.06–57.39%). However, Bacteroidetes in AOA_S150_ (26.17–40.33%) and AOA_S300_ (32.07–48.06%) reactors was significantly higher than it was in the MBR_control_ (11.03–12.51%) at the final stage of operation. Bacteria of the phylum of Bacteroidetes have the ability to grow via subsisting on the organism released through cell lysis [17,54]. Therefore, bacteria of the phylum Bacteroidetes probably contributed to sludge reduction. Additionally, the relative abundance of Planctomycetes, the anammox bacteria that are related to organics’ degradation [12,55], was higher in AOA_S300_ (4.13–7.85%) than in AOA_S150_ (1.72–3.93%) and the MBR_control_ (2.60–3.48%).

To determine the specific differences in microbial community between reactors, the bacterial composition was also analyzed at the genus level. *Dechloromonas*, *Zoogloea*, *Ignavibacterium*, and SRB were the dominant microbes in the three bioreactors during the operation. As shown in Figure 4b, the MBR_control_ exhibited the highest *Dechloromonas* proportion on day 90, at 11.98–31.04%. Meanwhile, in AOA_S150_ and AOA_S300_, the relative abundance of *Dechloromonas* decreased to 2.58–12.65% and 4.73–27.37%, respectively. However, *Zoogloea*, the aerobic denitrifying bacterium that also plays a role in high-molecular-weight compound removal [32,56,57], was enriched in AOA_S150_ (1.16–4.92%) and AOA_S300_ (5.38–6.96%) after 90 days of operation, as compared with the MBR_control_ (0.29–0.62%).

Furthermore, AOAS_150_ and AOA_S300_ had higher abundances of *Ignavibacterium* compared with the MBR_control_ (lower than 0.1% on day 90). The relative abundance of *Ignavibacterium* in the AOAS_150_ bioreactor varied from 5.29 to 9.41% (day 1) and from 13.06 to 18.00% (day 90). During the whole operation, the abundance of *Ignavibacterium* in the AOAS_300_ bioreactor was the highest, reaching 16.93–30.88%. *Ignavibacterium* is the sulfur autotrophic denitrifying bacteria [58]. The sulfate added to the membrane region of AOA reactors could be reduced to S^2−^, which could be used as the electron donor for simultaneous nitrogen and sulfur consumption [59]. SRB, including *Desulfocapsa*, *Desulfovirga*, *Desulfovibrio*, *Desulforhabdus*, *Desulfonema*, *Desulfomonile*, *Desulfomicrobium*, *Desulfobu*, *Desulfobacterium*, *Desulfobacter*, and *Desulfatirhabdium*, are prokaryotes that have the capacity to use organic matter as electronic donors to reduce sulfate to sulfide under anaerobic conditions [48,60,61,62]. The relative abundance of SRB in AOA_S150_ (1.11–4.56%) and AOA_S300_ (0.96–1.52%) was higher than in the MBR_control_ (0.26–0.40%), indicating that the sulfate reduction in AOA_S150_ and AOA_S300_ was more active, which probably induced the higher consumption of organic matter and sludge reduction. Huang et al. [18] operated SBR_AO_, SBR_OSA_, and SBR_SOSA_ reactors (adding sulfate in the OSA tank) and found that, compared with SBR_AO_, the SBR_SOSA_ process could reduce the sludge yield by 57%. However, sulfate-reducing bacteria were only abundant in the OSA tank of SBR_SOSA_ (2.1%), while they were rare in other tanks (0.1–0.7%), since sulfate-reducing bacteria were easily affected by oxygen and nitrate in the environment.

To identify the possible correlation among different genera in the three bioreactors, a network analysis was performed by using Gephi (v 0.9.2) [63], with a Pearson’s correlation coefficient ρ > 0.6 and significance *p* < 0.01 (Appendix A). Nodes with identical colors were classified into the same module, and the node size was determined by the relative abundance of the genus. The red and green connection lines represent positive and negative correlations, respectively [64]. As shown in Appendix A, there were more positive correlations than negative correlations. The sulfate-reducing bacterium *Desulfovibrio* was positively related with *Chloroherpeton* and *Chlorobaculum*, the green sulfur bacteria [65,66]. *Desulfocapsa* and *Desulfomicrobium* (SRB) had a positive correlation with the green sulfur bacterium *Chloroherpeton*, hydrolytic denitrifying bacterium *Flavobacterium* [13,67], and green sulfur bacterium *Chlorobaculum*, respectively. In addition, *Ignavibacterium*, the sulfur autotrophic denitrification bacterium, correlated with fermentative bacterium *Longilinea Anaerolinea* [14,68] and green sulfur bacterium *Chlorobaculum* positively. Therefore, bacteria related to sulfur metabolism and hydrolytic fermentative bacteria interacted to promote the sludge reduction in the system.

### 3.5. Metabolic Response Pathways

The sludge reduction process was not only associated with the microbial community structure; it was also closely related to the metabolic mechanism of microorganisms in the system. Analyzing the distribution of functional genes in the bioreactors could provide in-depth information about the sludge reduction process.

#### 3.5.1. Glycolysis Pathway

Glycolysis could effectively transform the six-carbon glucose molecule into two three-carbon pyruvate molecules [69]. Figure 5 compared the relative abundance of important enzyme-encoding genes during glucose degradation. The relative abundance of glucokinase (EC: 2.7.1.2) and polyphosphate glucokinase (EC: 2.7.1.63), which were involved in the initial glycolysis step in the AOA_S150_ and AOA_S300_ bioreactors, were 0.0163% and 0.0160% (day 1), 0.0103% and 0.0175% (day 45), and 0.0139% and 0.0148% (day 90), respectively. The total abundance of these two genes in the AOA_S300_ bioreactor was higher than in the AOA_S150_ bioreactor, which probably had a direct effect on the subsequent glycolysis process and, thus, affected the electron-generation process [70]. Phosphofructokinase (EC: 2.7.1.11) is a critical enzyme that determines the overall rate of the glycolysis reaction [69]. The relative abundances of the functional genes encoding this enzyme in the AOA_S150_ and AOA_S300_ bioreactors were 0.0237% and 0.0242% (day 1), 0.0232% and 0.0219% (day 45), and 0.0267% and 0.0217% (day 90), respectively. Phosphoglycerate kinase (EC: 2.7.2.3) and pyruvate kinase (EC: 2.7.1.40) are key enzymes catalyzing ATP generation [71]. The total relative abundance of genes encoding these two enzymes in the AOAS_150_ bioreactor decreased from 0.0347% (day 1) to 0.0251% (day 45) and then increased to 0.0391% (day 90). In the AOAS_300_ bioreactor, that decreased from 0.0303% (day 1) to 0.0281% (day 90). In the AOA_S150_ bioreactor, the total relative abundance of functional genes involved in the entire glycolysis process decreased from 0.161% (day 1) to 0.128% (day 45) and then increased to 0.156% (day 90). Meanwhile, in the AOA_S300_ bioreactor, the total relative abundance of functional genes involved in the whole glycolysis process gradually decreased from 0.150% (day 1) to 0.130% (day 90). It could be speculated that microorganisms in the AOA_S300_ bioreactor may be gradually inhibited, contributing to the decreased sludge generation [71,72].

#### 3.5.2. Nitrogen Metabolic Pathway

The glycolysis process could generate ATP and NADH for microbial nitrogen metabolism [73]. The relative abundances of enzyme-encoding genes related to nitrogen metabolism are shown in Figure 6. The total relative abundances of enzyme-encoding genes involved in nitrification in the AOA_S150_ and AOA_S300_ bioreactors were 0.014% and 0.019% (day 1), and 0.012% and 0.019% (day 45), respectively. However, at the final stage of operation, the total relative abundances of enzyme-encoding genes involved in nitrification in the AOA_S150_ and AOA_S300_ bioreactors were 0.036% and 0.011%, respectively, thus indicating that nitrification may be inhibited in the AOA_S300_ reactor [74]. During this period, the total abundance of enzyme-encoding genes involved in the denitrifying process in the AOA_S150_ bioreactor increased from 0.051% (day 1) to 0.082% (day 90), while it slightly decreased from 0.054% (day 1) to 0.053% (day 90) in the AOA_S300_ bioreactor. Therefore, the denitrification potential of the AOA_S300_ bioreactor might be inhibited, possibly due to the accumulation of sulfide, which might decrease the generation of sludge [48].

#### 3.5.3. Sulfate Reduction Pathway

To further investigate the sulfate reduction metabolism in the AOA_S150_ and AOA_S300_ bioreactors, the enzyme-encoding genes involved in the sulfate-reduction metabolism in the membrane tanks of the AOA_S150_ and AOA_S300_ bioreactors were analyzed, and the metabolism pathway was plotted, as shown in Figure 7.

Sulfate can be transformed to sulfide through assimilation and dissimilation reduction [75]. There are four steps in assimilatory sulfate reduction: sulfate → adenosine phosphosulfate (APS) → 3′-phosphoadenylyl sulfate (PAPS) → sulfite → sulfide. The dissimilatory sulfate-reduction pathway consists of three steps: sulfate → APS → sulfite → sulfide [76]. The total relative abundances of enzyme-encoding genes involved in sulfate reduction in the AOA_S150_ bioreactor were 0.047% (day 1), 0.059% (day 45), and 0.072% (day 90), respectively. The total relative abundances of enzyme-encoding genes in the AOA_S300_ bioreactor decreased from 0.058% (day 1) to 0.042% (day 90). The sulfate-reduction potential of the AOA_S300_ bioreactor was likely to decrease gradually, which could be due to the sulfide produced by sulfate reduction accumulated in the AOA_S300_ bioreactor and gradually inhibited SRB activity [77]. Sulfide could accelerate the sludge decay rate [18], and sulfide inhibition might contribute to sludge reduction.

## 4. Conclusions

Sulfidogenic AOA bioreactors achieved efficient sludge minimization through sulfide-induced sludge lysis and EPS destruction. The sludge yields in AOA bioreactors were 40–50% lower than those of the MBR_control_, and nitrogen removal was stimulated due to sulfur autotrophic denitrification. In AOA_S150_, a gradual decrease in protein-like EPS was recorded from the pre-anoxic to the membrane tank, while in AOA_S300_, aerobic lysis happened when the sludge flowed from the pre-anoxic to the oxic tank, and the organics released were consumed in the membrane tank. Bacteroidetes (phylum), SRB, and *Ignavibacterium* (genus) were enriched in AOA bioreactors compared with the MBR_control_. The metagenomic analysis suggested that the accumulation of sulfide in AOA_S300_ inactivates the functional bacteria, and sulfide inhibition could contribute to sludge reduction. This study can provide important guidance for the operational regulation of AOA sludge-reduction processes.

## Figures and Tables

**Figure 1 membranes-12-00865-f001:**
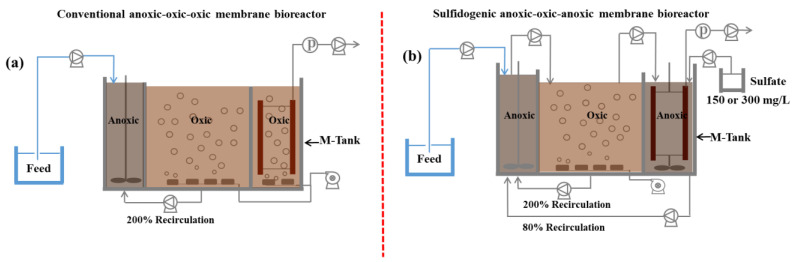
Configuration of conventional (**a**) membrane bioreactor (MBR_control_) and (**b**) sulfidogenic anoxic–oxic–anoxic membrane bioreactors (AOA_S150_ and AOA_S300_).

**Figure 2 membranes-12-00865-f002:**
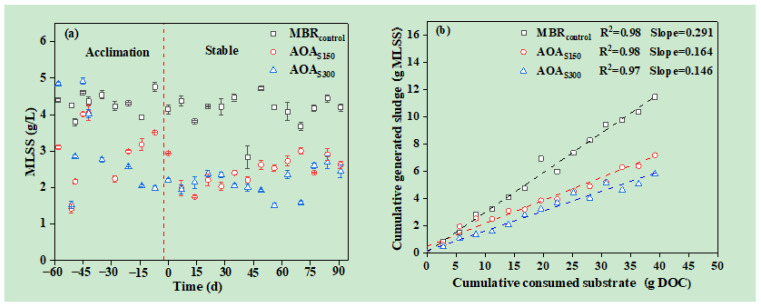
Change in the average MLSS concentration between acclimation and stable operation phases (**a**) and (**b**) the observed sludge yield (Y_obs_) in the MBR_control_, AOA_S150_, and AOA_S300_ bioreactors.

**Figure 3 membranes-12-00865-f003:**
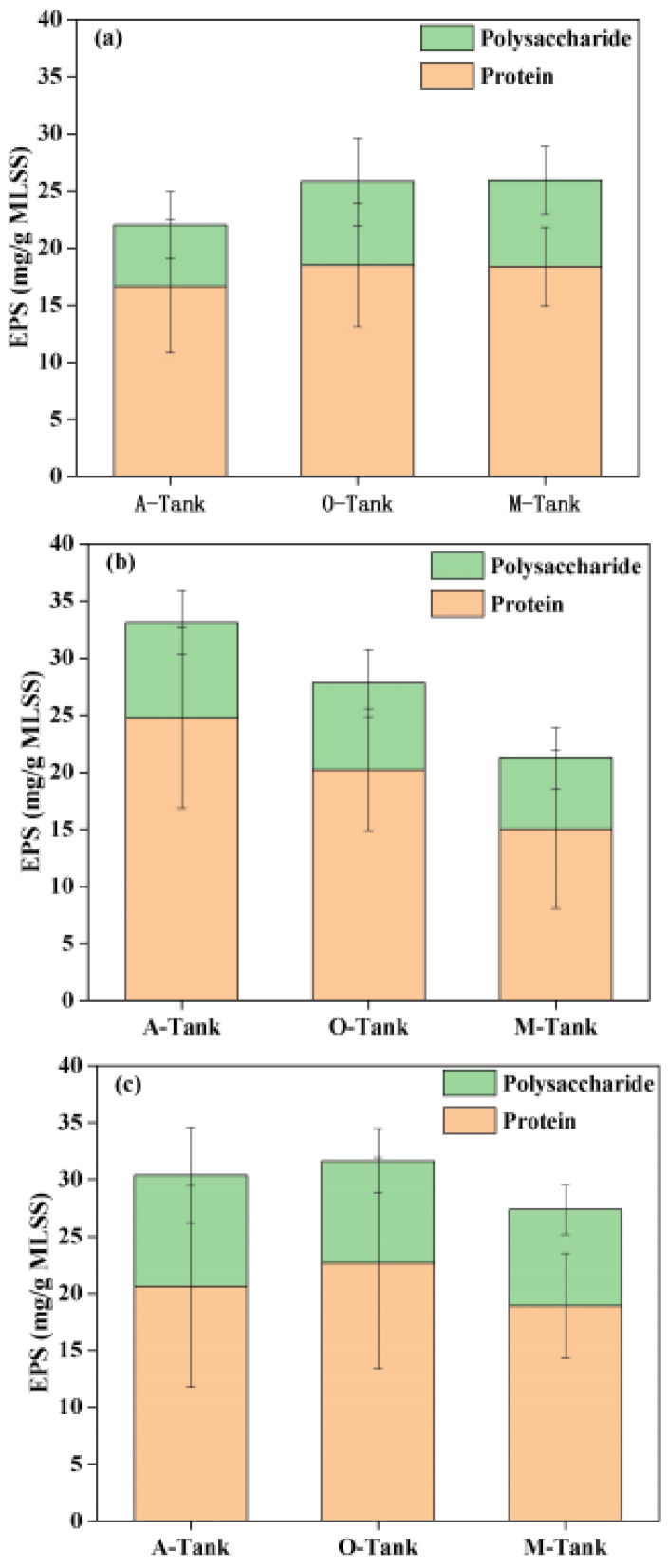
Total EPS in terms of polysaccharides and protein in different tanks of three bioreactors, MBR_control_ (**a**), AOA_S150_ (**b**), and AOA_S300_ (**c**).

**Figure 4 membranes-12-00865-f004:**
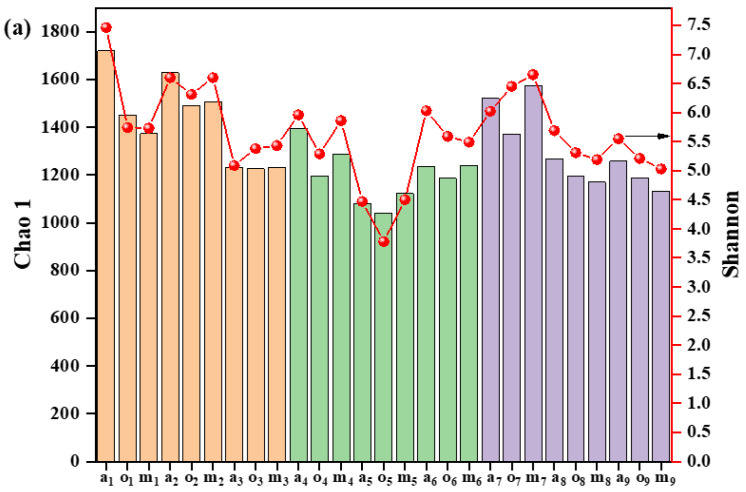
(**a**) Chao 1 and Shannon (red line) indexes, relative abundance of dominant species at (**b**) the phylum and (**c**) the genus level. (Here, a, o, and m are on behalf of the anoxic, oxic, and membrane tank, respectively. In detail, a_1_–m_3_, a_4_–m_6_, and a_7_–m_9_ mean the samples of MBR_control_, AOA_S150_, and AOA_S300_. The subscript number denotes the sampling time. For example, 1, 4, and 7 represent the initial stage of MBR_control_, AOA_S150_, and AOA_S300_; 2, 5, and 8 represent the middle stage; and 3, 6, and 9 represent the final stage of MBR_control_, AOA_S150_, and AOA_S300_, respectively).

**Figure 5 membranes-12-00865-f005:**
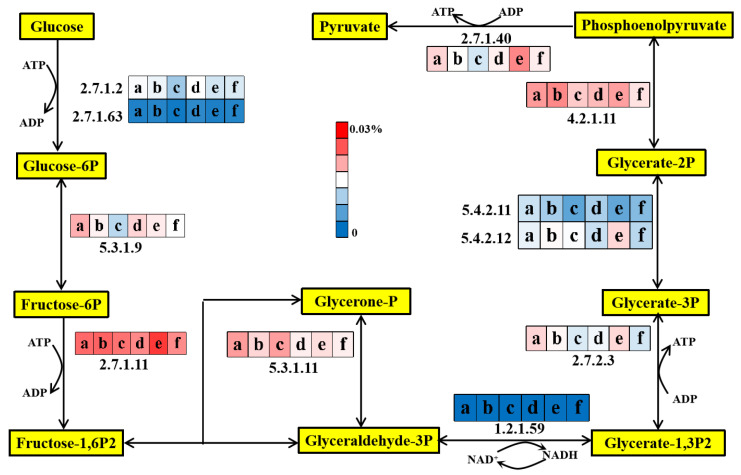
Glycolysis pathway and average relative abundances of enzyme-encoding genes in AOA_S150_ and AOA_S300_ bioreactors (a, c, and e represent the initial, middle, and final stage of AOA_S150_; b, d, and f represent the initial, middle, and final stage of AOA_S300_, respectively).

**Figure 6 membranes-12-00865-f006:**
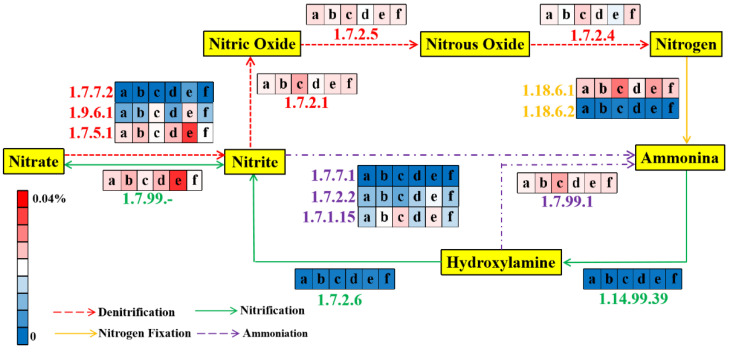
Nitrogen pathway and average relative abundances of enzyme-encoding genes in AOA_S150_ and AOA_S300_ bioreactors (a, c, and e represent the initial, middle, and final stage of AOA_S150_; b, d, and f represent the initial, middle, and final stage of AOA_S300_, respectively).

**Figure 7 membranes-12-00865-f007:**
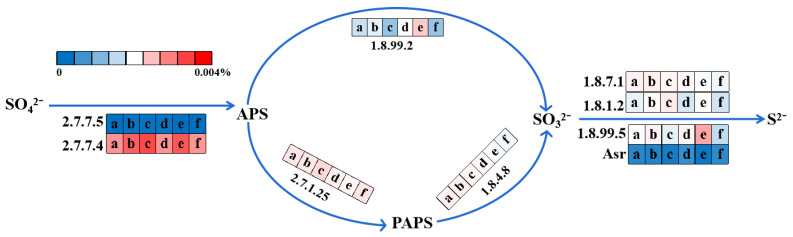
Sulfate reduction pathway and average relative abundances of enzyme-encoding genes in AOA_S150_ and AOA_S300_ bioreactors (a, c, and e represent the initial, middle, and final stage of AOA_S150_; b, d, and f represent the initial, middle, and final stage of AOA_S300_, respectively).

## Data Availability

All data are available in the main text or Appendix A.

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
