# Peer review of "In-Situ Sludge Reduction Performance and Mechanism in Sulfidogenic Anoxic–Oxic–Anoxic Membrane Bioreactors"

_membranes, 2022, doi:10.3390/membranes12090865_

Round 1

Reviewer 1 Report

The authors investigated on the In-situ sludge reduction performance and mechanism in sulfidogenic anoxic–oxic–anoxic membrane bioreactors. They operated three bioreactors with different sulfate concentration. The authors evaluated the process performance and microbial community structures. In my opinion, this paper seems to be interesting, and is well written. I recommend minor revision with follow comments.

1. KEGG facilitates to guess the metabolic pathway, but it cannot confirm the pathway. However, in current manuscript, the authors address confirmed metabolic pathway. I think that the authors should revise these expressions throughout the manuscript.

2. What is primary mechanism of using sulfide for sludge reduction in this study? Please clearly suggest.

3. How the authors evaluated metabolic response pathways? Please address the methodology in detail.

Reviewer 2 Report

In this manuscript, the sludge reduction performance of sulfidogenic AOA membrane bioreactors was investigated, and the dynamics of the microbial community and metabolic pathways were further analyzed. Generally, it shows some interesting results on sludge reduction of MBR assisted with sulfidogenic bioaugmentation. However, there are some questions as listed below. I can not recommend acceptance of this manuscript before addressing these questions.

(1)    Introduction section, the abbreviations such as AO and AOA should be defined during their first appearance.

(2)   For MBRs, not nm but µm is usually used as the unit of the pore size of MF or UF membrane.

(3)   In this manuscript, the water flux of MBRs was controlled at 5 LMH. Compared with a typical MBR, the flux value was much lower. What is the reason?

(4)   I strongly suggest summarizing the effluent water quality of three MBRs in a table for better comparing the performance of different MBRs.

(5)   Everybody knows that the membrane performance is important to evaluate the operating state of MBR. However, the authors did not show the data on the flux performance and TMP variation of the three MBRs.

(6)   It is well known that sludge properties had a significant impact on membrane fouling. In this manuscript, owing to the sludge reduction, the sludge properties such as sludge concentration had a dramatic change. In this case, the membrane fouling in MBR might be significantly influenced. Unfortunately, the authors did not evaluate the membrane fouling in the three MBRs.

Round 2

Reviewer 2 Report

The authors have done major revisions to the manuscript. In my opinion, the authors have addressed the comments raised by the reviewers, and the quality of the manuscript has been improved. Thus, I recommend acceptance of the revised manuscript.